# Hepatic NF-kB-inducing kinase (NIK) suppresses mouse liver regeneration in acute and chronic liver diseases

Yi Xiong[1†], Adriana Souza Torsoni[1,2†], Feihua Wu[1,3], Hong Shen[1], Yan Liu[1], Xiao Zhong[1], Mark J Canet[1], Yatrik M Shah[1], M Bishr Omary[1], Yong Liu[4], Liangyou Rui[1,5*]

[1]Department of Molecular and Integrative Physiology, University of Michigan Medical School, Ann Arbor, United States; [2]Laboratory of Metabolic Disorders, School of Applied Sciences, University of Campinas, Limeira, Brazil; [3]Department of Pharmacology of Chinese Materia Medica, School of Traditional Chinese Medicine, China Pharmaceutical University, Nanjing, China; [4]College of Life Sciences, Institute for Advanced Studies, Wuhan University, Wuhan, China; [5]Department of Internal Medicine, University of Michigan Medical School, Ann Arbor, United States

**Abstract** Reparative hepatocyte replication is impaired in chronic liver disease, contributing to disease progression; however, the underlying mechanism remains elusive. Here, we identify Map3k14 (also known as NIK) and its substrate Chuk (also called IKKα) as unrecognized suppressors of hepatocyte replication. Chronic liver disease is associated with aberrant activation of hepatic NIK pathways. We found that hepatocyte-specific deletion of *Map3k14* or *Chuk* substantially accelerated mouse hepatocyte proliferation and liver regeneration following partial-hepatectomy. Hepatotoxin treatment or high fat diet feeding inhibited the ability of partial-hepatectomy to stimulate hepatocyte replication; remarkably, inactivation of hepatic NIK markedly increased reparative hepatocyte proliferation under these liver disease conditions. Mechanistically, NIK and IKKα suppressed the mitogenic JAK2/STAT3 pathway, thereby inhibiting cell cycle progression. Our data suggest that hepatic NIK and IKKα act as rheostats for liver regeneration by restraining overgrowth. Pathological activation of hepatic NIK or IKKα likely blocks hepatocyte replication, contributing to liver disease progression.
DOI: https://doi.org/10.7554/eLife.34152.001

**\*For correspondence:**
ruily@umich.edu

[†]These authors contributed equally to this work

**Competing interests:** The authors declare that no competing interests exist.

## Introduction

The liver is an essential metabolic organ that experiences metabolic stress during fasting, refeeding, and overnutrition states (*Rui, 2014*). The liver is also responsible for detoxifications of endogenous and exogenous toxic substances, thus being frequently exposed to harmful insults. Dietary hepatotoxins and gut microbiota-derived toxic substances are transported to the liver through the enterohepatic circulation, further increasing risk for liver injury. To compensate for hepatocyte loss, the liver evolves a powerful regenerative ability to maintain its homeostasis (*Michalopoulos, 2017*). After 70% of partial hepatectomy (PHx), rodents are able to regain normal liver mass within a week via reparative hepatocyte replications (*Miyaoka et al., 2012*). Nevertheless, hepatocyte proliferation is severely inhibited in chronic liver diseases, including nonalcoholic fatty liver disease (NAFLD), alcoholic liver disease, and chronic exposures to hepatotoxins (*Richardson et al., 2007*; *Inaba et al., 2015*; *Sancho-Bru et al., 2012*; *Michalopoulos, 2013*). Impairment in hepatocyte replications considerably precipitates liver disease progression; however, the underlying mechanism responsible for defective hepatocyte replications remains poorly understood.

In response to liver injury induced by PHx, numerous growth factors and cytokines are secreted and delivered to hepatocytes where they stimulate hepatocyte proliferation by activating multiple mitogenic pathways, including the Janus kinase 2 (JAK2)/STAT3, MAPK, PI 3-kinase, and NF-kB pathways (*Michalopoulos, 2017*). In contrast, TGFβ1 and interferon-γ inhibit hepatocyte proliferation, thereby preventing liver from overgrowth (*Michalopoulos, 2013*; *Sato et al., 1993*; *Wu et al., 2015*). Liver regeneration is fine-tuned by a balance between positive and negative regulators. We postulated that in chronic liver disease, the negative branch might be predominant and overcome the positive branch, leading to pathological suppression of hepatocyte proliferation and liver regeneration. However, intracellular pathways conferring hepatocyte proliferation inhibition remain elusive. In search for inhibitory pathways, we identified Map3k14, also called NF-kB-inducing kinase (NIK), and its substrate Chuck, also referred to as IkB kinase α (IKKα).

NIK is a Ser/Thr kinase known to activate the noncanonical NF-kB2 pathway (*Sun, 2012*). It phosphorylates and activates IKKα (*Xiao et al., 2001*). IKKα in turn phosphorylates the precursor of NF-kB2 p100, resulting in generation of the p52 form of NF-kB2 (*Sun, 2012*; *Xiao et al., 2001*). Mature p52 is translocated into the nucleus to activate target genes. We previously reported that metabolic stress, oxidative stress, hepatotoxins, and cytokines stimulate hepatic NIK (*Sheng et al., 2012*; *Jiang et al., 2015*). Importantly, hepatic NIK is aberrantly activated in both mice and humans with NAFLD or alcoholic liver disease (*Sheng et al., 2012*; *Shen et al., 2014*). Hepatocellular stress and liver inflammation, which are associated with chronic liver disease, likely activate hepatic NIK. These observations prompted us to test the hypothesis that hepatic NIK/IKKα pathways cell-autonomously inhibit hepatocyte proliferation. In this work, we characterized hepatocyte-specific NIK (NIK$^{\Delta hep}$) and IKKα (IKKα$^{\Delta hep}$) knockout mice, and examined reparative hepatocyte replications using PHx models. We found that the NIK/IKKα pathway suppresses reparative hepatocyte proliferation at least in part by inhibiting the JAK2/STAT3 pathway. This work unveils unrecognized crosstalk between the NIK/IKKα and the JAK2/STAT3 pathways involved in regulating liver regeneration.

## Results

### Hepatocyte-specific ablation of NIK accelerates liver regeneration

To assess the role of hepatic NIK in reparative hepatocyte proliferation, we performed 70% of PHx on mice at 8 weeks of age following the established protocols (*Mitchell and Willenbring, 2008*). NIK$^{\Delta hep}$ mice were generated by crossing *Map3k14$^{flox/flox}$* (referred to as NIK$^{f/f}$) mice with *Albumin-Cre* drivers (*Shen et al., 2017*). Proliferating cells were detected by immunostaining liver sections with antibody against Ki67, a cell proliferation marker (*Figure 1A*). Baseline hepatocyte proliferation rates were low and comparable between NIK$^{\Delta hep}$ and NIK$^{f/f}$ mice (*Figure 1B*). Number of liver proliferating Ki67$^+$ cells markedly increased within 48 hr following PHx, and Ki67$^+$ cells were 85% higher in NIK$^{\Delta hep}$ relative to NIK$^{f/f}$ mice (*Figure 1B*). In line with these observations, the number of liver BrdU-labelled proliferating cells was also substantially higher in NIK$^{\Delta hep}$ than in NIK$^{f/f}$ mice (*Figure 1C*). Liver cell proliferation rates declined in both NIK$^{\Delta hep}$ and NIK$^{f/f}$ mice after 48 hr post-PHx, and became comparable between these two groups at 96 hr post-PHx (*Figure 1B*).

To verify hepatocytes proliferating, we costained liver sections with anti-Ki67 and anti-HNF4α (a hepatocyte marker) antibodies, or with anti-Ki67 and anti-F4/80 (a Kupffer cell/macrophage marker) antibodies. HNF4α$^+$ hepatocytes accounted for 96% of Ki67$^+$ proliferating cells in NIK$^{\Delta hep}$ mice at 48 hr post-PHx (*Figure 1D,F*) while F4/80$^+$ Kupffer cells/macrophages for <4% of Ki67$^+$ cells (*Figure 1E,F*). These data indicate that hepatic NIK is an intrinsic suppressor of hepatocyte proliferation.

We also examined the effect of NIK deficiency on hepatocyte death using TUNEL assays. The number of liver TUNEL$^+$ apoptotic cells was slightly lower in NIK$^{\Delta hep}$ relative to NIK$^{f/f}$ mice, but the difference was not statistically significant (*Figure 1G*). Plasma alanine aminotransferase (ALT) activity, a liver injury index, was comparable between NIK$^{\Delta hep}$ and NIK$^{f/f}$ mice either under basal conditions or after PHx (*Figure 1H*). Thus, accelerated hepatocyte proliferation cannot be explained by changes in liver injury in NIK$^{\Delta hep}$ mice.

To further confirm the role of hepatic NIK in liver regeneration, we assessed liver to body weight ratios at 2 and 4 days post-PHx. Consistently, liver/body weight ratios were significantly higher in NIK$^{\Delta hep}$ than in NIK$^{f/f}$ mice at 4 days following PHx (*Figure 1I*). Of note, liver/body weight ratios

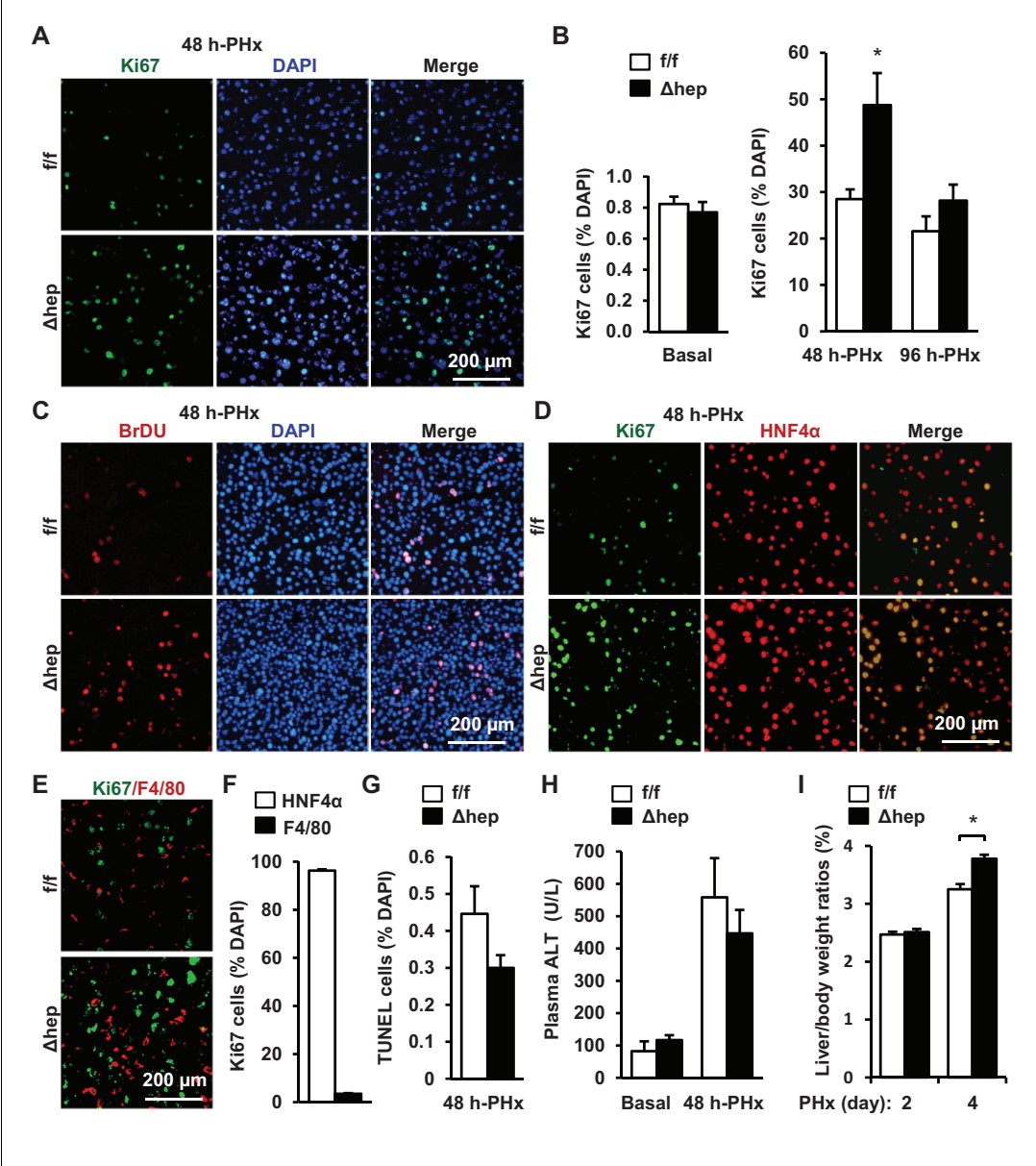

**Figure 1.** Hepatocyte-specific ablation of NIK accelerates reparative hepatocyte proliferation. NIK[f/f] (n = 7) and NIK[Δhep] (n = 7) male mice (8 weeks) were subjected to PHx, and livers were harvested 48 hr or 96 hr later. (**A**) Representative immunostaining of liver sections (48 hr after PHx) with anti-Ki67. (**B**) Ki67[+] cells were counted and normalized to total DAPI[+] cells. (**C**) Representative immunostaining of liver sections (48 hr after PHx) with anti-BrdU antibodies. (**D–E**) Representative images of liver sections (48 hr after PHx) costained with anti-Ki67 and anti-HNF4α antibodies (**D**) or anti-Ki67 and anti-F4/80 antibodies (**E**). (**F**) Ki67[+]HNF4α[+] and Ki67[+]F4/80[+] cells were counted and normalized to total Ki67[+] cells. (**G**) Liver cell death were assessed 48 hr after PHx using TUNEL reagents. (**H**) Plasma ALT levels. (**I**) Liver to body weight ratios (n = 8 per group). Data were statistically analyzed with two-tailed Student's t test, and presented as mean ± SEM. *p<0.05.

DOI: https://doi.org/10.7554/eLife.34152.002

The following source data is available for figure 1:

**Source data 1.** Hepatic NIK deficiency accelerates liver regeneration.
DOI: https://doi.org/10.7554/eLife.34152.003
**Source data 2.** PHx increases hepatocyte replications.
DOI: https://doi.org/10.7554/eLife.34152.004

were similar between these two groups at 2 days post-PHx. One possible explanation is that a 2 day period may be too short for newly-generated hepatocytes to grow in size large enough to increase liver weight.

To determine whether NIK inhibits hepatocyte cell cycle progression, we measured the levels of cyclin D1, which is believed to drive hepatocyte proliferation following PHx (*Michalopoulos, 2013*). Hepatic cyclin D1 levels were undetectable in both NIK$^{\Delta hep}$ and NIK$^{f/f}$ mice under basal conditions, and were markedly increased by PHx (*Figure 2A*). Importantly, hepatic cyclin D1 levels were significantly higher in NIK$^{\Delta hep}$ than in NIK$^{f/f}$ mice (*Figure 2A,B*). Collectively, these results support the notion that hepatic NIK may act as an intrinsic rheostat for liver homeostasis by restraining liver overgrowth.

## The role of NF-kB1, MAPK, and PI 3-kinase pathways in NIK-induced suppression of hepatocyte proliferation

We next sought to interrogate the molecular mechanism of the NIK action. Expression of liver NIK rapidly increased within 12 hr following PHx, but declined at 3 days post-PHx (*Figure 2—figure supplement 1A*). Consistently, PHx also increased phosphorylation of liver IKKα/β (*Figure 2—figure supplement 1B,C*). Interestingly, liver IKKα expression was also increased by PHx (*Figure 2—figure supplement 1C*). The NF-kB1, MAPK, and PI 3-kinase pathways are known to be involved in mediating PHx-stimulated liver regeneration (*Michalopoulos, 2013*; *Wuestefeld et al., 2013*; *Pauta et al., 2016*). Unexpectedly, phosphorylation of hepatic IkBα, p65 (the NF-kB1 pathway), Akt (pSer473) (the PI 3-kinase pathway), ERK1/2, and JNK (the MAPK pathway) was comparable between NIK$^{\Delta hep}$ and NIK$^{f/f}$ mice at 4 hr post-PHx (*Figure 2C*). We also did not detect difference in hepatic levels of reactive oxygen species (ROS) or hepatic expression of cytokines between NIK$^{\Delta hep}$ and NIK$^{f/f}$ mice (*Figure 2D,E*). Therefore, NIK suppression of liver regeneration cannot be explained by the above pathways.

## NIK suppresses the JAK2/STAT3 pathway

JAK2 is known to phosphorylate and activate STAT3, which is believed to drive hepatocyte proliferation (*Wang et al., 2011*; *Shi et al., 2017*). We postulated that NIK might suppress hepatocyte proliferation by inhibiting the JAK2/STAT3 pathway. Liver extracts were prepared at 4 hr post-PHx and immunoblotted with anti-phospho-JAK2 (pTyr1007/1008) or anti-phospho-STAT3 (pTyr705) antibodies. Phosphorylation of both JAK2 and STAT3 was significantly higher in NIK$^{\Delta hep}$ mice than in NIK$^{f/f}$ littermates (*Figure 3A*). Baseline levels of JAK2 and STAT3 phosphorylation in the resected livers were similar between NIK$^{\Delta hep}$ and NIK$^{f/f}$ mice (*Figure 2—figure supplement 1D*).

To confirm that NIK directly inhibits the JAK2/STAT3 pathway, we transiently coexpressed JAK2 and STAT3 with NIK in HEK293 cells. In line with our previous reports (*Rui and Carter-Su, 1999*), overexpressed JAK2 robustly autophosphorylated as well as phosphorylated STAT3 (*Figure 3B*). Strikingly, overexpression of NIK dramatically decreased tyrosine phosphorylation of both JAK2 and STAT3 (*Figure 3B*). Consistently, NIK was coimmunoprecipitated with JAK2 (*Figure 3C*). These data indicate that NIK binds to JAK2 and inhibits JAK2 activity, thereby suppressing the JAK2/STAT3 pathway.

Interleukin 6 (IL6) stimulates the JAK2/STAT3 pathway, which is required for reparative hepatocyte proliferation (*Riehle et al., 2008*; *Cressman et al., 1996*). These observations prompted us to test if NIK negatively regulates the IL6/JAK2/STAT3 pathway. Mouse primary hepatocytes were transduced with NIK or β-galactosidase (β-gal) adenoviral vectors, followed by IL6 stimulation. IL6 robustly stimulated phosphorylation of STAT3 in β-gal-transduced, but not NIK-transduced, hepatocytes (*Figure 3D*). Collectively, these results unveil unrecognized crosstalk between NIK pathways and the JAK2/STAT3 pathway.

## Hepatic IKKα suppresses liver regeneration following PHx

Given that IKKα acts downstream of NIK in the noncanonical NF-kB2 pathway, we reasoned that hepatic IKKα might also suppress liver regeneration. IKKα$^{\Delta hep}$ mice were generated by crossing *Chuk$^{flox/flox}$* (referred to as IKKα$^{f/f}$) mice with *albumin-Cre* drivers (*Liu et al., 2008*). We confirmed that IKKα expression was disrupted specifically in the liver but not brain, heart, kidney, skeletal muscle, and spleen in IKKα$^{\Delta hep}$ mice (*Figure 4A*). We performed PHx on IKKα$^{f/f}$ and IKKα$^{\Delta hep}$ male mice

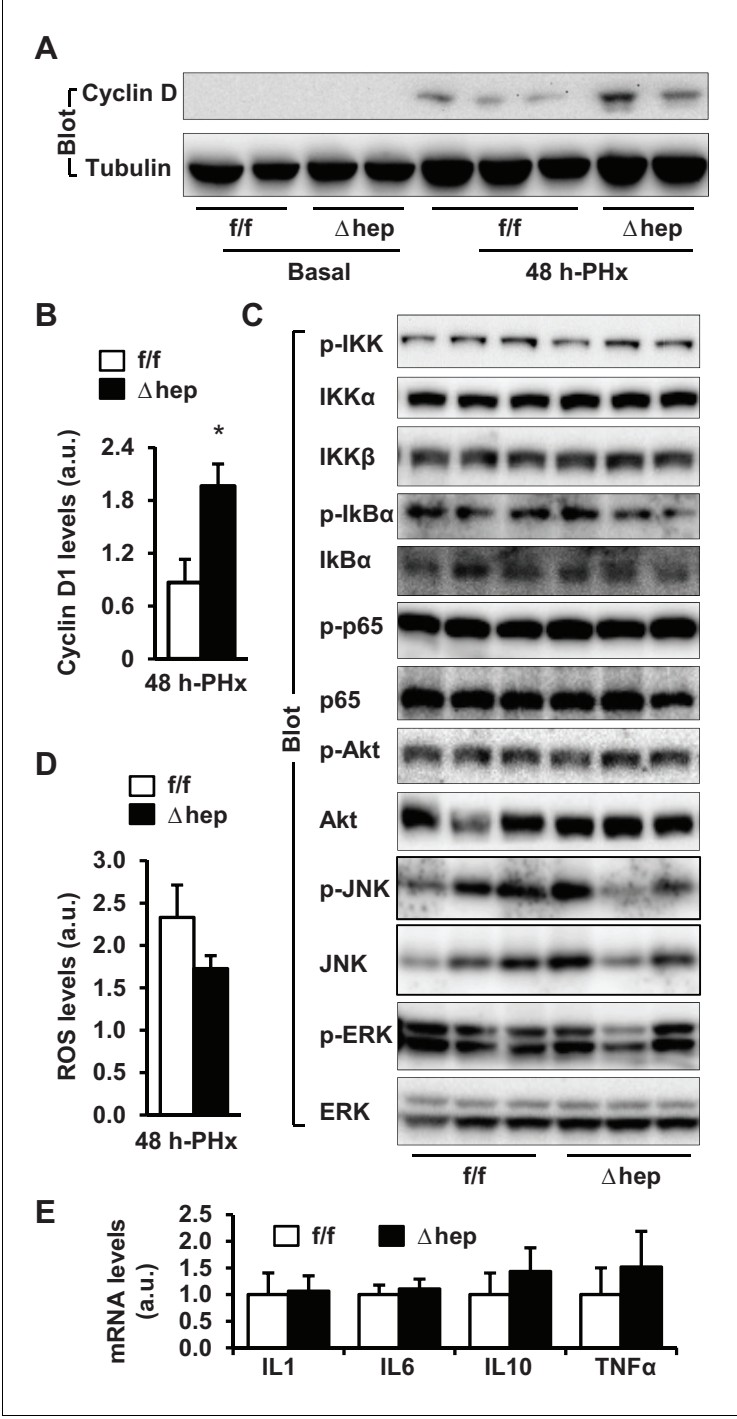

**Figure 2.** Hepatic NIK deficiency upregulates cyclin D1 without altering NF-kB1, Akt, and MAPK pathways in the liver. NIK[f/f] and NIK[Δhep] male mice (8 weeks) were subjected to PHx. (**A–B**) Liver extracts were immunoblotted with anti-cyclin D1 antibody (48 hr after PHx). Cyclin D1 levels were quantified and normalized to α-tubulin levels (NIK[f/f]: n = 4, NIK[Δhep]: n = 4). (**C**) Liver extracts were immunoblotted with the indicated antibodies (4 hr after PHx). (**D**) Liver ROS levels 48 hr after PHx (normalized to liver weight). NIK[f/f]: n = 5, NIK[Δhep]: n = 6. (**E**) Liver cytokine expression was measured by qPCR and normalized to 36B4 expression (48 hr after PHx). NIK[f/f]: n = 5, NIK[Δhep]: n = 5. Data were statistically analyzed with two-tailed Student's t test, and presented as mean ± SEM. *p<0.05.
DOI: https://doi.org/10.7554/eLife.34152.005

The following source data and figure supplements are available for figure 2:

**Source data 1.** Hepatic NIK regulates hepatocyte cell cycle progression.

*Figure 2 continued on next page*

*Figure 2 continued*

DOI: https://doi.org/10.7554/eLife.34152.008

**Figure supplement 1.** Effect of PHx on liver NIK pathway activation.

DOI: https://doi.org/10.7554/eLife.34152.006

**Figure supplement 1—source data 1.** PHx stimulates hepatic NIK expression.

DOI: https://doi.org/10.7554/eLife.34152.007

at 8–9 weeks of age. The number of liver proliferating Ki67[+] cells was significantly higher in IKKα[Δhep] than in IKKα[f/f] littermates at both 1 and 2 days post-PHx, and became similar between these two groups after 3 days following PHx (*Figure 4B*). HNF4α[+] hepatocytes accounted for the majority of proliferating cells (*Figure 4C*). Consistently, liver cyclin D1 levels were significantly higher in IKKα[Δhep] than in IKKα[f/f] mice (*Figure 4D*), while liver cell death was comparable between these two groups (*Figure 4E*). Consequently, liver to body weight ratios were significantly higher in IKKα[Δhep] relative to IKKα[f/f] mice at both 5 and 7 days post-PHx (*Figure 4F*). Notably, liver/body weight ratios were comparable between these two groups within 3 days following PHx, likely due to lack of sufficient time for hepatocytes to grow their mass as discussed before. These results indicate that deficiency

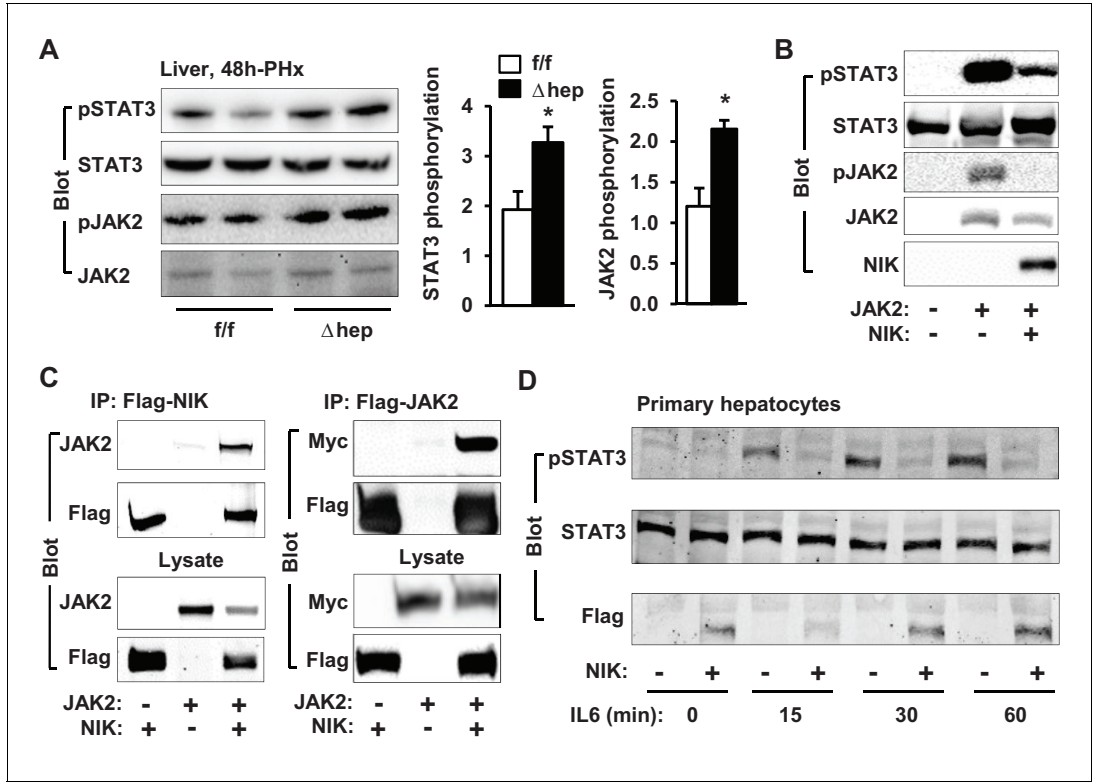

**Figure 3.** NIK inhibits the JAK2/STAT3 pathway. (A) Liver extracts were prepared from NIK[f/f] and NIK[Δhep] males 4 hr after PHx and immunoblotted with anti-phospho-JAK2 and anti-phospho-STAT3 antibodies. Phosphorylation of JAK2 (pTyr1007/1008) and STAT3 (pTyr705) was normalized to total JAK2 and STAT3 levels, respectively. (B) STAT3 and JAK2 were coexpressed with or without NIK in HEK293 cells. Cell extracts were immunoblotted with the indicated antibodies. (C) NIK was coexpressed with JAK2 in HEK293 cells. Cell extracts were immunoprecipitated (IP) and immunoblotted with the indicated antibodies. (D) Mouse primary hepatocytes were transduced with NIK or β-gal adenoviral vectors and stimulated with IL6 (10 ng/ml). Cell extracts were immunoblotted with the indicated antibodies. Data were statistically analyzed with two-tailed Student's t test, and presented as mean ± SEM. *p<0.05.

DOI: https://doi.org/10.7554/eLife.34152.009

The following source data is available for figure 3:

**Source data 1.** NIK inhibits the JAK2/STAT3 pathway.

DOI: https://doi.org/10.7554/eLife.34152.010

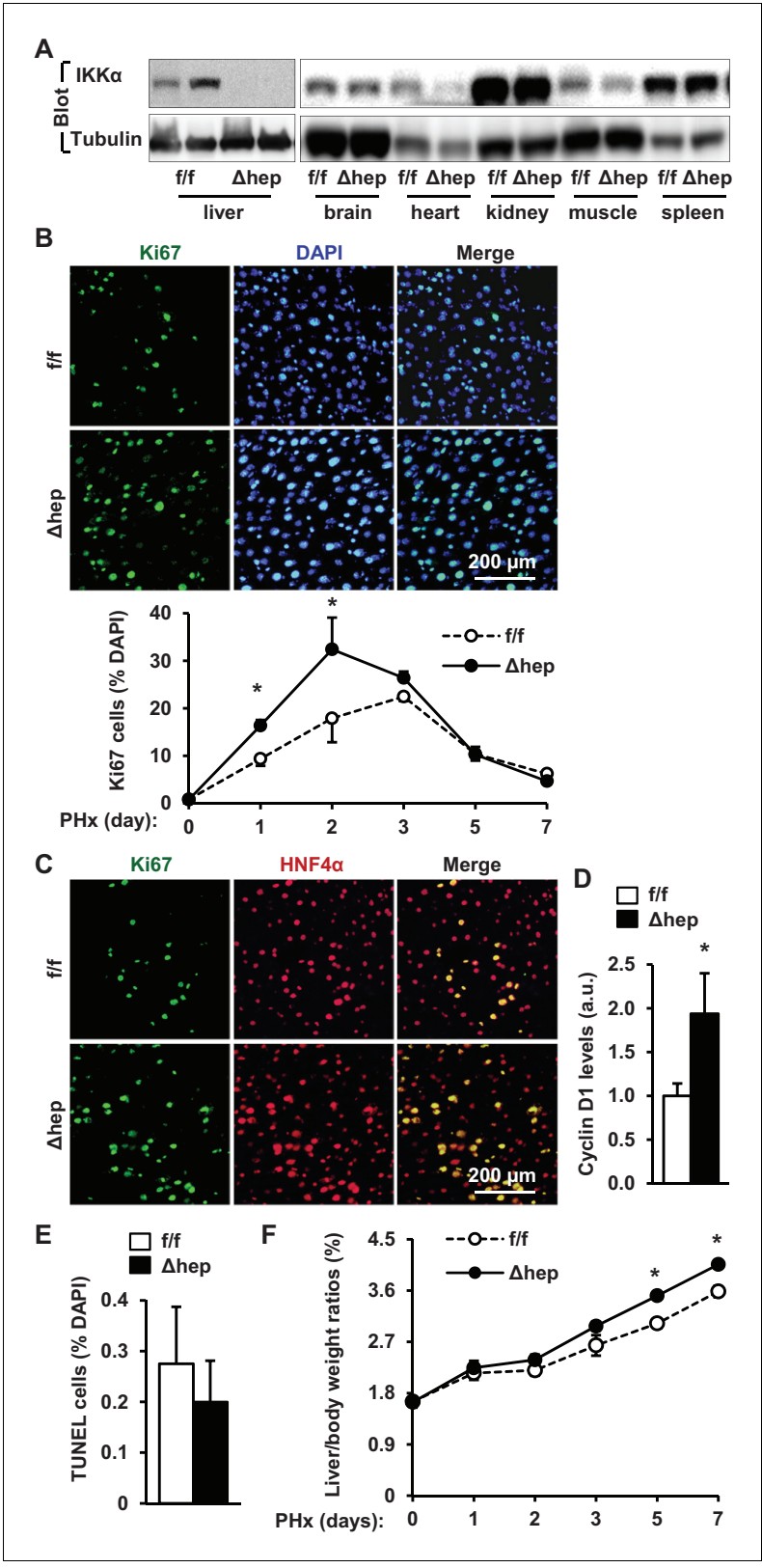

**Figure 4.** Ablation of hepatocyte IKKα accelerates hepatocyte reparative proliferation. (**A**) Tissue extracts were immunoblotted with anti-IKKα or anti-α-tubulin antibodies. (**B–F**) IKKα$^{f/f}$ (n = 6) and IKKα$^{\Delta hep}$ (n = 6) male littermates were subjected to PHx, and livers were harvested 48 hr later. (**B**) Liver sections were immunostained with anti-Ki67 antibody, and Ki67$^+$ cells were counted and normalized to total DAPI$^+$ cells. Day 0 and 1: n = 4 per

*Figure 4 continued on next page*

*Figure 4 continued*

group; day 3: IKKα^f/f: n = 6, IKKα^Δhep: n = 8; day 5: IKKα^f/f: n = 9, IKKα^Δhep: n = 8; day 7: IKKα^f/f: n = 6, IKKα^Δhep: n = 5. (C) Representative images of liver sections costained with anti-Ki67 and anti-HNF4α antibodies. (D) Liver cyclin D1 was measured by immunoblotting (normalized to α-tubulin levels). (E) TUNEL-positive cells in liver sections. (F) Liver to body weight ratios. Day 0 and 1: n = 4 per group; day 3: IKKα^f/f: n = 6, IKKα^Δhep: n = 8; day 5: IKKα^f/f: n = 9, IKKα^Δhep: n = 8; day 7: IKKα^f/f: n = 6, IKKα^Δhep: n = 5. Data were statistically analyzed with two-tailed Student's t test, and presented as mean ± SEM. *p<0.05.

DOI: https://doi.org/10.7554/eLife.34152.011

The following source data is available for figure 4:

**Source data 1.** Hepatic IKKα regulates liver regeneration.
DOI: https://doi.org/10.7554/eLife.34152.012

---

of hepatocyte IKKα, like NIK, also accelerates hepatocyte proliferation and liver regeneration in response to acute liver injury.

To gain insight into the molecular mechanism of the IKKα action, we examined the JAK2/STAT3 pathway. The levels of phosphorylation of JAK2 as well as STAT3 were significantly higher in IKKα^Δhep than in IKKα^f/f mice at 4 hr post-PHx (*Figure 5A,B*). We also compared phosphorylation time courses during days 0–7 following PHx. IKKα phosphorylation increased while JAK2 phosphorylation decreasing during days 1–5 (*Figure 5—figure supplement 1A,B*). This inverse relationship further supports the notion that the NIK/IKKα pathway inhibits the JAK2/STAT3 pathway. Ablation of hepatocyte IKKα increased phosphorylation of JAK2 and STAT3 during days 1–7 following PHx (*Figure 5—figure supplement 1B*). To confirm that IKKα cell-autonomously inhibits the JAK2/STAT3 pathway, IKKα was transiently coexpressed with JAK2 in HEK293 cells. IKKα was coimmunoprecipitated with JAK2 (*Figure 5C*), and markedly decreased JAK2 autophosphorylation and the ability of JAK2 to phosphorylate STAT3 (*Figure 5D*).

To determine whether NIK suppresses the JAK2/STAT3 pathway via IKKα, we transduced primary hepatocytes from IKKα^Δhep (IKKα-deficient) and IKKα^f/f (wild type) mice with NIK or green fluorescent protein (GFP) adenoviral vectors, followed by IL6 stimulation. The ability of NIK to inhibit IL6-stimulated phosphorylation of STAT3 was significantly reduced in IKKα-deficient hepatocytes compared to wild type hepatocytes (*Figure 5—figure supplement 1C,D*). Of note, NIK overexpression still considerably attenuated STAT3 phosphorylation in IL6-stimulated IKKα-deficient hepatocytes, compared with GFP overexpression (*Figure 5—figure supplement 1D*). These findings suggest that hepatic NIK suppresses the JAK2/STAT3 pathway, and possibly liver regeneration, by both IKKα-dependent and IKKα-independent mechanisms.

## Deficiency of hepatic NIK accelerates liver regeneration in mice with hepatotoxin-induced liver injury

Hepatic NIK is highly activated in mice and humans with chronic liver disease (*Sheng et al., 2012*; *Shen et al., 2014*), raising the possibility that hepatic NIK might impair liver regeneration in these disease conditions. To model chronic liver disease, we treated NIK^Δhep and NIK^f/f male mice with 2-acetylaminofluorene (AAF), a hepatotoxin (*Laishes and Rolfe, 1981*), for 10 days prior to PHx. Liver cell proliferation was assessed at 48 hr post-PHx. AAF treatment considerably increased hepatic levels of NF-kB2 p52 in wild type mice, indicative of NIK activation (*Figure 6A*). Baseline levels of proliferating Ki67^+ hepatocytes in the resected liver (<2%) were comparable between NIK^f/f and NIK^Δhep mice (*Figure 6B*). PHx markedly increased hepatocyte proliferation rates in NIK^f/f mice, which was substantially inhibited by AAF pretreatment (*Figure 6C,D*). Remarkably, the number of Ki67^+ hepatocytes was significantly higher in NIK^Δhep relative to NIK^f/f littermates following AAF and PHx treatments (*Figure 6C,D*). Liver to body weigh ratios were slightly higher in NIK^Δhep relative to NIK^f/f mice at 2 days post-PHx, but not statistically different (*Figure 6—figure supplement 1A*). As discussed above, a 2 day period may be too short for newly-generated hepatocytes to grow in size to significantly increase liver weight. Plasma ALT levels were also similar between NIK^f/f and NIK^Δhep mice (*Figure 6E*).

We next examined cell signaling that drives cell cycle progression. We detected baseline levels of phosphorylation of hepatic STAT3 in NIK^Δhep but not NIK^f/f mice after AAF pretreatment (*Figure 6F*). PHx stimulated STAT3 phosphorylation in both NIK^Δhep and NIK^f/f mice, but to a substantially higher

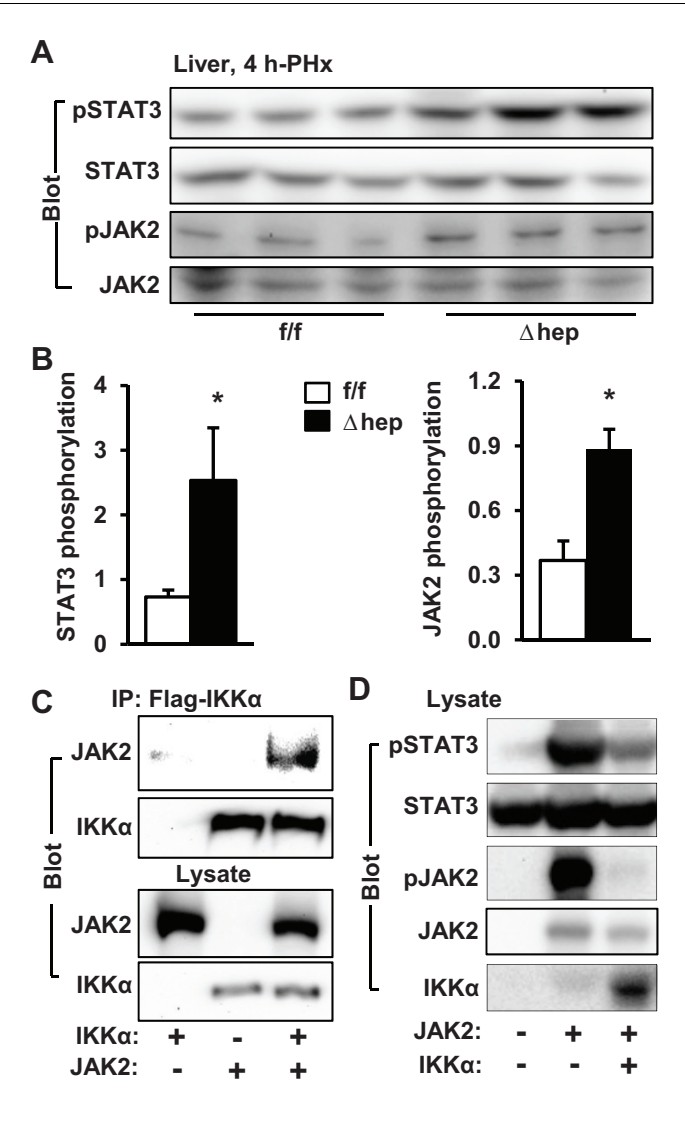

**Figure 5.** IKKα inhibits the JAK2/STAT3 pathway. (A–B) Liver extracts were prepared 4 hr after PHx and immunoblotted with anti-phospho-JAK2 and anti-phospho-STAT3 antibodies. Phosphorylation of JAK2 (pTyr1007/ 1008) and STAT3 (pTyr705) was normalized to total JAK2 and STAT3 levels, respectively. IKKα$^{f/f}$: n = 6, IKKα$^{\Delta hep}$: n = 6. (C) IKKα and JAK2 were coexpressed in HEK293 cells. Cell extracts were immunoprecipitated (IP) and immunoblotted with the indicated antibodies. (D) STAT3 and JAK2 were coexpressed with IKKα in HEK293 cells. Cell extracts were immunoblotted with the indicated antibodies. Data were statistically analyzed with two-tailed Student's t test, and presented as mean ± SEM. *p<0.05.

DOI: https://doi.org/10.7554/eLife.34152.013

The following source data and figure supplements are available for figure 5:

**Source data 1.** IKKα regulates the JAK2/STAT3 pathway.
DOI: https://doi.org/10.7554/eLife.34152.017

**Figure supplement 1.** The effect of PHx on activation of liver IKKα and JAK2/STAT3 pathways.
DOI: https://doi.org/10.7554/eLife.34152.014

**Figure supplement 1—source data 1.** PHx regulates the hepatic JAK2/STAT3 pathway.
DOI: https://doi.org/10.7554/eLife.34152.015

**Figure supplement 1—source data 2.** Regulation of the JAK2/STAT3 pathway by NIK/IKKα pathways.
DOI: https://doi.org/10.7554/eLife.34152.016

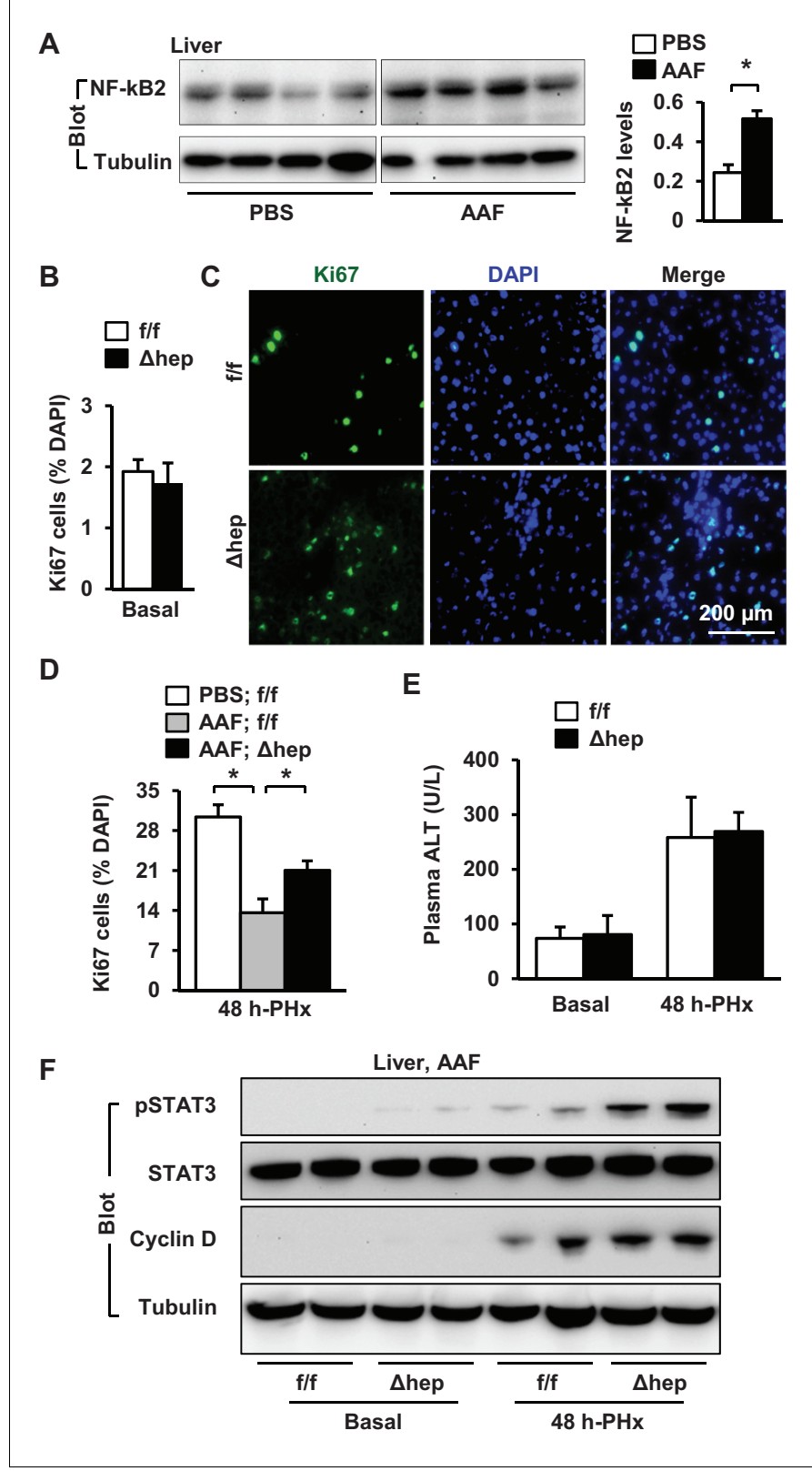

**Figure 6.** Ablation of hepatocyte NIK reverses AAF-induced impairment in hepatocyte reparative proliferation. (**A**) C57BL/6 males (8 weeks) were treated with PBS or AAF (10 mg/kg body weight, gavage) daily for 10 days. NF-kB2 p52 in liver extracts was immunoblotted with anti-NF-kB2 antibody (normalized to α-tubulin levels). PBS: n = 4, AAF: n = 4. (**B–G**) NIK$^{f/f}$ and NIK$^{Δhep}$ males were treated with PBS or AAF (10 mg/kg body weight) for 10 days and

*Figure 6 continued on next page*

*Figure 6 continued*

then subjected to PHx. Livers were harvested 48 hr later. (**B**) Baseline Ki67$^+$ cell number in resected liver sections obtained from PHx. NIK$^{f/f}$: n = 5, NIK$^{\Delta hep}$: n = 4. (**C**) Representative immunostaining of liver sections (AAF-treated) with anti-Ki67 antibody. (**D**) Ki67$^+$ cell number in liver sections (normalized to DAPI$^+$ cells). PBS;NIK$^{f/f}$: n = 3, AAF; NIK$^{f/f}$: n = 5, AAF;NIK$^{\Delta hep}$: n = 5. (**E**) Plasma ALT levels. NIK$^{f/f}$: n = 3, NIK$^{\Delta hep}$: n = 4. (**F**) Liver extracts were immunoblotted with the indicated antibodies. Data were statistically analyzed with two-tailed Student's t test, and presented as mean ± SEM. *p<0.05.

DOI: https://doi.org/10.7554/eLife.34152.018

The following source data and figure supplements are available for figure 6:

**Source data 1.** Hepatic NIK regulates hepatocyte proliferation in AAF-treated mice.
DOI: https://doi.org/10.7554/eLife.34152.022
**Figure supplement 1.** Hepatic NIK inhibits reparative hepatocyte proliferation.
DOI: https://doi.org/10.7554/eLife.34152.019
**Figure supplement 1—source data 1.** Baseline hepatocyte proliferation in AAF-treated mice.
DOI: https://doi.org/10.7554/eLife.34152.020
**Figure supplement 1—source data 2.** Baseline hepatocyte proliferation in HFD-fed mice.
DOI: https://doi.org/10.7554/eLife.34152.021

level in NIK$^{\Delta hep}$ mice (*Figure 6F*). Baseline hepatic cyclin D1 levels were undetectable in both NIK$^{\Delta hep}$ and NIK$^{f/f}$ mice pretreated with AAF, and PHx increased cyclin D1 levels to a higher extent in NIK$^{\Delta hep}$ than in NIK$^{f/f}$ mice (*Figure 6F*). Together, these data support the notion that abnormal activation of hepatic NIK contributes to hepatotoxin-induced impairment in liver regeneration.

## Inactivation of hepatic NIK increases reparative hepatocyte proliferation in mice with NAFLD

NAFLD is associated with both arrest of hepatocyte proliferation and upregulation of hepatic NIK (*Richardson et al., 2007*; *Inaba et al., 2015*; *Sheng et al., 2012*; *Shen et al., 2014*; *Collin de l'Hortet et al., 2014*), prompting us to test if elevated hepatic NIK is responsible for impairment in liver regeneration under the disease conditions. To model NAFLD, we placed NIK$^{\Delta hep}$ and NIK$^{f/f}$ mice on a high fat diet (HFD) for 10 weeks. Both NIK$^{\Delta hep}$ and NIK$^{f/f}$ mice similarly developed liver steatosis, as assessed by liver triacylglycerol (TAG) levels (*Figure 7A*). HFD feeding increased hepatic NF-kB2 p52 levels, indicative of NIK activation (*Figure 7B*). To assess liver regeneration, we performed PHx after HFD feeding for 10 weeks. Hepatocyte proliferation was assessed at 48 hr post-PHx by staining liver sections with anti-Ki67 antibody (*Figure 7C*). Baseline levels of hepatocyte proliferation in the resected liver were comparable between NIK$^{\Delta hep}$ and NIK$^{f/f}$ mice (*Figure 7D*). PHx markedly increased hepatocyte proliferation in chow-fed NIK$^{f/f}$ mice, which was substantially inhibited by HFD feeding (*Figure 7E*). Importantly, number of proliferating Ki67$^+$ hepatocytes was significantly higher in NIK$^{\Delta hep}$ than in NIK$^{f/f}$ littermates after HFD feeding (*Figure 7E*). Liver/body weight ratios were slightly higher in NIK$^{\Delta hep}$ relative to NIK$^{f/f}$ mice at 2 days post-PHx, but not statistically different (*Figure 6—figure supplement 1B*). This modest difference can be explained by the short duration that limits the capacity of newly-generated hepatocytes to significantly grow in size and increase liver weight. Plasma ALT levels were comparable between NIK$^{\Delta hep}$ and NIK$^{f/f}$ littermates under both basal and PHx conditions (*Figure 7F*).

We further explored liver mitogenic pathways in these mice. Baseline STAT3 phosphorylation levels in the resected liver were similar between NIK$^{\Delta hep}$ and NIK$^{f/f}$ mice fed HFD; however, liver STAT3 phosphorylation increased to a considerably higher level in NIK$^{\Delta hep}$ relative to NIK$^{f/f}$ mice at 48 hr post-PHx (*Figure 7G*). Hepatic cyclin D1 levels were also higher in NIK$^{\Delta hep}$ than in NIK$^{f/f}$ mice post-PHx (*Figure 7G*). These data suggest that aberrant activation of hepatic NIK suppresses hepatocyte proliferation and liver regeneration in NAFLD at least in part by inhibiting the JAK2/STAT3 pathway.

## Discussion

Reparative hepatocyte proliferation plays a pivotal role in the maintenance of liver homeostasis and integrity by supplying new hepatocytes to replace lost ones. Liver regeneration impairment is likely involved in chronic liver disease. In this work, we identified hepatic NIK and IKKα as unrecognized

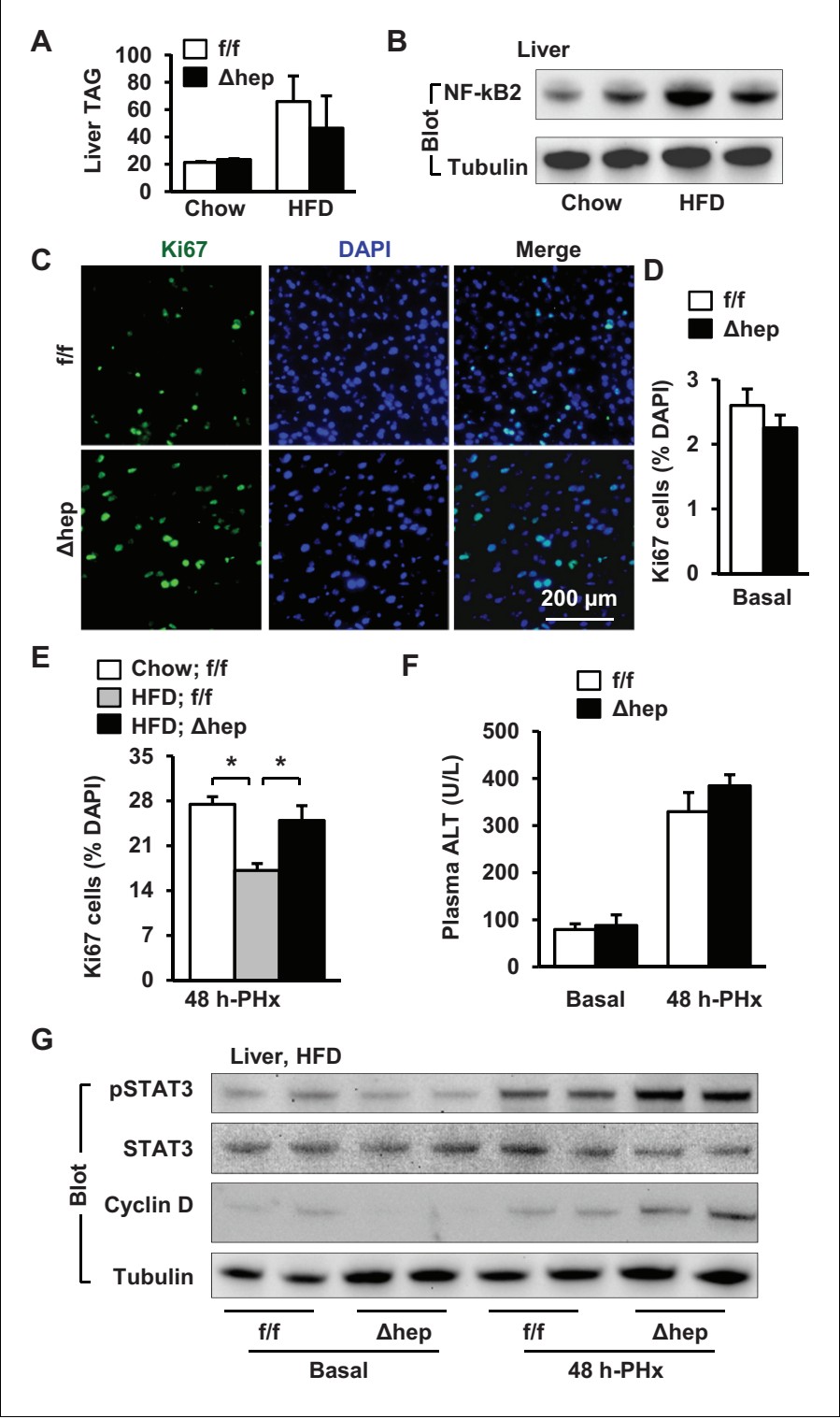

**Figure 7.** Hepatic NIK deficiency corrects impaired hepatocyte reparative proliferation in mice with NAFLD. (**A–B**) C57BL/6 males (8 weeks) were fed a normal chow diet (n = 5) or a HFD (n = 5) for 10 weeks. (**A**) Liver TAG levels (normalized to liver weight). (**B**) NF-kB2 p52 in liver extracts was immunoblotted with anti-NF-kB2 antibody (normalized to $\alpha$-tubulin levels). (**C–H**) NIK[f/f] and NIK[Δhep] males were fed a HFD for 10 weeks followed by PHx, and livers were harvested 48 hr after PHx. (**C**) Representative immunostaining of liver sections with anti-Ki67 antibody. (**D**) Baseline Ki67[+] cell number in resected liver sections obtained from PHx. NIK[f/f]: n = 4, NIK[Δhep]: n = 4. (**E**) Liver Ki67[+] cell number (normalized to DAPI[+] cells). Chow;NIK[f/f]: n = 3, HFD;NIK[f/f]: n = 5, HFD; NIK[Δhep]: n = 4. (**F**)

*Figure 7 continued on next page*

*Figure 7 continued*

Plasma ALT levels. NIK$^{f/f}$: n = 3, NIK$^{\Delta hep}$: n = 4. (**G**) Liver extracts were immunoblotted with the indicated antibodies. Data were statistically analyzed with two-tailed Student's t test, and presented as mean ± SEM. *p<0.05.

DOI: https://doi.org/10.7554/eLife.34152.023

The following source data is available for figure 7:

**Source data 1.** Hepatic NIK regulates hepatocyte proliferation in HFD-fed mice.

DOI: https://doi.org/10.7554/eLife.34152.024

suppressors of liver regeneration; moreover, NIK inhibits hepatocyte proliferation at least in part by activating IKKα. We previously demonstrated that hepatic NIK is aberrantly activated in mice and humans with chronic liver disease (*Sheng et al., 2012*; *Shen et al., 2014*). Our current results show that elevated activation of hepatic NIK pathways impairs liver regeneration, likely contributing to liver disease progression.

We found that hepatocyte-specific ablation of NIK or IKKα substantially increases hepatocyte proliferation in NIK$^{\Delta hep}$ or IKKα$^{\Delta hep}$ mice following PHx. Accordingly, liver regeneration rates were higher both in NIK$^{\Delta hep}$ relative to NIK$^{f/f}$ littermates and in IKKα$^{\Delta hep}$ relative to IKKα$^{f/f}$ mice. We observed that both NIK and IKKα bound to JAK2 and substantially inhibited the ability of JAK2 to phosphorylate STAT3. Consistently, hepatocyte-specific ablation of either NIK or IKKα substantially increased phosphorylation of hepatic JAK2 and STAT3 in mice post-PHx. IKKα deficiency decreased the ability of NIK to suppress the JAK2/STAT3 pathway in hepatocytes, confirming that IKKα acts downstream of NIK. However, NIK overexpression still inhibited the JAK2/STAT3 pathway in IKKα-deficient hepatocytes, suggesting that hepatic NIK is able to suppress the JAK2/STAT3 pathway by an additional IKKα-independent mechanism. The JAK2/STAT3 pathway is known to drive hepatocyte proliferation, which is indispensable for liver regeneration (*Wang et al., 2011*; *Shi et al., 2017*; *Riehle et al., 2008*; *Cressman et al., 1996*). Therefore, hepatic NIK and IKKα inhibit liver regeneration at least in part by suppressing the JAK2/STAT3 pathway.

Mounting evidence shows that hepatic NIK is aberrantly activated in chronic liver disease, likely due to liver inflammation and hepatocellular stress (*Sheng et al., 2012*; *Shen et al., 2014*). We modeled chronic liver disease by chronically treating mice with hepatotoxin AAF or placing them on HFD. We found that hepatocyte-specific inactivation of NIK substantially increases the ability of PHx to stimulate hepatocyte proliferation in both AAF-treated mice and HFD-fed NIK$^{\Delta hep}$ mice. Consistently, in mice pretreated with AAF or HFD, ablation of hepatic NIK increased phosphorylation of both hepatic JAK2 and STAT3 post-PHx. It is worth mentioning that NIK in nonparenchymal cells (e.g. immune cells) also contributes to obesity-associated liver steatosis (*Liu et al., 2017*). These observations raise the possibility that in chronic liver disease, NIK in Kupffer cells/macrophages, and possibly other nonparenchymal cells, may indirectly inhibit reparative hepatocyte replication by a paracrine mechanism. Collectively, our results provide proof of concept evidence supporting the notion that aberrant hepatic NIK impairs reparative hepatocyte replication, thereby contributing to liver disease progression.

In conclusion, we have identified hepatic NIK and IKKα as unrecognized suppressors of reparative hepatocyte replication. NIK and IKKα suppress liver regeneration at least in part by inhibiting the hepatic JAK2/STAT3 pathway. Our findings suggest that pharmacological inhibition of hepatic NIK or IKKα may provide a new therapeutic strategy for liver disease treatment.

# Materials and methods

**Key resources table**

| Reagent type | Designation | Source or reference | Identifiers | Additional information |
|---|---|---|---|---|
| Antibody | Ki67 | Vector lab | VP-RM04 | 1:100 |
| Antibody | NIK | Abcam | ab191591 | 1:2000 |
| Antibody | IKK beta | Cell Signaling Technology | 8943 | 1:5000 |

*Continued on next page*

*Continued*

| Reagent type | Designation | Source or reference | Identifiers | Additional information |
|---|---|---|---|---|
| Antibody | IKK alpha | Cell Signaling Technology | 2682 | 1:5000 |
| Antibody | p-IKKa/b | Cell Signaling Technology | 2697 | 1:5000 |
| Antibody | STAT3 | Santa Cruz | sc-8019 | 1:1000 |
| Antibody | p-STAT3 | Cell Signaling Technology | 9145 | 1:5000 |
| Antibody | JAK2 | Cell Signaling Technology | 3230 | 1:5000 |
| Antibody | p-JAK2 1007/1008 | Cell Signaling Technology | 3776 | 1:5000 |
| Antibody | Myc | Santa Cruz | sc-40 | 1:1000 |
| Antibody | Flag | Sigma | F1804 | 1:5000 |
| Antibody | p85 | Home-raised | N/A | 1:5000 |
| Antibody | $\alpha$-tubulin | Santa Cruz | sc-5286 | 1:1000 |
| Antibody | JNK | Cell Signaling Technology | 9258 | 1:5000 |
| Antibody | p-JNK | Cell Signaling Technology | 4668 | 1:5000 |
| Antibody | ERK1/2 | Cell Signaling Technology | 9102 | 1:5000 |
| Antibody | p-ERK1/2 | Cell Signaling Technology | 4370 | 1:5000 |
| Antibody | NF-kB2 | Cell Signaling Technology | 4882 | 1:5000 |
| Antibody | p65 | Cell Signaling Technology | 8242 | 1:5000 |
| Antibody | p-p65 | Cell Signaling Technology | 3033 | 1:5000 |
| Antibody | IkB alpha | Cell Signaling Technology | 4812 | 1:5000 |
| Antibody | p-IkB alpha | Cell Signaling Technology | 9246 | 1:5000 |
| Antibody | AKT | Cell Signaling Technology | 4091 | 1:5000 |
| Antibody | p-Akt | Cell Signaling Technology | 4060 | 1:5000 |
| Antibody | Cyclin D1 | Cell Signaling Technology | 2978 | 1:5000 |
| Antibody | F4/80 | eBioscience | 14–4801 | 1:100 |
| Antibody | HNF4 alpha | Santa Cruz | sc-8987 | 1:100 |
| Antibody | CK8 | Developmental Studies Hybridoma Bank | Troma I | 1:100 |
| Antibody | BrDU | Cell Signaling Technology | 5292 | 1:100 |

## Antibodies and animals

Antibodies were described in the key resources table. Animal experiments were conducted following the protocols approved by the University of Michigan Institutional Animal Care and Use Committee (IACUC). We generated NIK$^{f/f}$, NIK$^{\Delta hep}$, and IKK$\alpha^{\Delta hep}$ mice (C57BL/6 background). IKK$\alpha^{f/f}$ mice (C57BL/6 background) were provided by Dr. Yinling Hu (the Inflammation and Tumorigenesis Section, National Cancer Institute). *Albumin-Cre* mice (C57BL/6 background) were from the Jackson laboratory (Bar Harbor, ME). Mice were housed on a 12 hr light-dark cycle and fed a normal chow diet (9% fat; Lab Diet, St. Louis, MO) or a HFD (60% fat in calories; D12492, Research Diets, New Brunswick, NJ) *ad libitum* with free access to water.

## PHx models

We followed published 2/3 PHx protocols (*Mitchell and Willenbring, 2008*). Briefly, NIK$^{f/f}$, NIK$^{\Delta hep}$, IKK$\alpha^{f/f}$, and IKK$\alpha^{\Delta hep}$ male mice (8–10 wks,) were anesthetized with isoflurane, followed by a ventral midline incision. The median and left lateral lobes (70% of the liver) were resected by pedicle ligations. Mice were euthanized after PHx, and tissues were harvested for histological and biochemical analyses. Mice were introperitoneally injected, 12 hr before euthanization, with BrdU (40 mg/kg body weight, ip) to label proliferating cells. A separate cohort was fed a HFD for 10 weeks prior to PHx. An additional cohort was treated with hepatotoxin 2-acetylaminofluorene (AAF) (10 mg/kg body weight, gavage) daily for 10 days prior to PHx.

### Immunostaining

Liver frozen sections were prepared using a Leica cryostat (Leica Biosystems Nussloch GmbH, Nussloch, Germany), fixed in 4% paraformaldehyde for 30 min, blocked for 3 hr with 5% normal goat serum (Life Technologies) supplemented with 1% BSA, and incubated with the indicated antibodies at 4°C overnight. The sections were incubated with Cy2 or Cy3-conjugated secondary antibodies.

### Cell cultures, transient transfection, and adenoviral transductions

Primary hepatocytes were prepared from mouse liver using type II collagenase (Worthington Biochem, Lakewood, NJ) and grown on William's medium E (Sigma) supplemented with 2% FBS, 100 units $ml^{-1}$ penicillin, and 100 µg $ml^{-1}$ streptomycin, and infected with adenoviruses as described previously (*Zhou et al., 2009*). HEK293 cells were grown at 37°C in 5% $CO_2$ in DMEM supplemented with 25 mM glucose, 100 U $ml^{-1}$ penicillin, 100 µg $ml^{-1}$ streptomycin, and 8% calf serum. For transient transfection, cells were split 16–20 hr before transfection. Expression plasmids were mixed with polyethylenimine (Sigma, St. Louis, MO) and introduced into cells. The total amount of plasmids was maintained constant by adding empty vectors. Cells were harvested 48 hr after transfection for biochemical analyses.

### Immunoprecipitation and immunoblotting

Cells or tissues were homogenized in a L-RIPA lysis buffer (50 mm Tris, pH 7.5, 1% Nonidet P-40, 150 mm NaCl, 2 mm EGTA, 1 mm $Na_3VO_4$, 100 mm NaF, 10 mm $Na_4P_2O_7$, 1 mm benzamidine, 10 µg $ml^{-1}$ aprotinin, 10 µg $ml^{-1}$ leupeptin, 1 mm phenylmethylsulfonyl fluoride). Tissue samples were homogenized in lysis buffer (50 mM Tris, pH 7.5, 1% Nonidet P-40, 150 mM NaCl, 2 mM EGTA, 1 mM $Na_3VO_4$, 100 mM NaF, 10 mM $Na_4P_2O_7$, 1 mM benzamidine, 10 µg/ml aprotinin, 10 µg/ml leupeptin; 1 mM phenylmethylsulfonyl fluoride). Proteins were separated by SDS-PAGE and immunoblotted with the indicated antibodies.

### Real-time quantitative PCR (qPCR) and ROS assays

Total RNAs were extracted using TRIzol reagents (Life technologies). Relative mRNA abundance of different genes was measured using SYBR Green PCR Master Mix (Life Technologies, 4367659). Liver lysates were mixed with a dichlorofluorescein diacetate fluorescent (DCF, Sigma, D6883) probe (5 µM) for 1 hr at 37°C. DCF fluorescence was measured using a BioTek Synergy 2 Multi-Mode Microplate Reader (485 nm excitation and 527 nm emission).

### Statistical analysis

Data were presented as means ± sem. Differences between two groups were analyzed using two-tailed Student's t tests. $p < 0.05$ was considered statistically significant.

## Acknowledgements

We thank Drs. Lin Jiang, Liang Sheng, Chengxin Sun, and Lei Yin and Michelle Jin for assistance and discussion. We thank Dr. Yinling Hu (Inflammation and Tumorigenesis Section, National Cancer Institute) for providing us with IKKα$^{f/f}$ mice. This study was supported by grants DK091591, DK115646, DK114220 (to LR) and DK47918 (to MBO) from the National Institutes of Health (NIH), fellowship #2013/07313–4 from São Paulo Research Foundation (FAPESP) (AST), and grant 81420108006 (to YL) from the National Natural Science Foundation of China. This work utilized the cores supported by the Michigan Diabetes Research and Training Center (NIH DK20572), the University of Michigan's Cancer Center (NIH CA46592), the University of Michigan Nathan Shock Center (NIH P30AG013283), and the University of Michigan Gut Peptide Research Center (NIH DK34933).

# Additional information

## Funding

| Funder | Grant reference number | Author |
|---|---|---|
| National Institute of Diabetes and Digestive and Kidney Diseases | DK091591 | Liangyou Rui |
| National Institute of Diabetes and Digestive and Kidney Diseases | DK114220 | Liangyou Rui |
| National Institute of Diabetes and Digestive and Kidney Diseases | DK115646 | Liangyou Rui |
| National Institute of Diabetes and Digestive and Kidney Diseases | DK47918 | M Bishr Omary |
| National Natural Science Foundation of China | 81420108006 | Yong Liu |
| Fundação de Amparo à Pesquisa do Estado de São Paulo | #2013/07313–4 | Yong Liu |

The funders had no role in study design, data collection and interpretation, or the decision to submit the work for publication.

## Author contributions

Yi Xiong, Adriana Souza Torsoni, Data curation, Formal analysis, Validation, Investigation, Methodology, Writing—review and editing; Feihua Wu, Hong Shen, Yan Liu, Mark J Canet, Data curation, Investigation, Writing—review and editing; Xiao Zhong, Data curation; Yatrik M Shah, M Bishr Omary, Resources, Investigation, Writing—review and editing; Yong Liu, Investigation, Writing—review and editing; Liangyou Rui, Conceptualization, Resources, Data curation, Formal analysis, Supervision, Funding acquisition, Validation, Investigation, Visualization, Methodology, Writing—original draft, Project administration, Writing—review and editing

## Author ORCIDs

Liangyou Rui (iD) http://orcid.org/0000-0001-8433-8137

## Ethics

Animal experimentation: This study was performed in strict accordance with the recommendations in the Guide for the Care and Use of Laboratory Animals of the National Institutes of Health. All of the animals were handled according to approved institutional animal care and use committee (IACUC) protocols (PRO00006638) of the University of Michigan. The protocol was approved by the Committee on the Ethics of Animal Experiments of the University of Michigan.

## Decision letter and Author response

Decision letter https://doi.org/10.7554/eLife.34152.027
Author response https://doi.org/10.7554/eLife.34152.028

# Additional files

## Supplementary files

• Transparent reporting form
DOI: https://doi.org/10.7554/eLife.34152.025

## Data availability

All data generated or analysed during this study are included in the manuscript and supporting files.

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
