## [Decision Letter]

Thank you for submitting your article "Hepatic NF-κB-inducing Kinase (NIK) Suppresses Liver Regeneration in Chronic Liver Disease" for consideration by *eLife*. Your article has been reviewed by three peer reviewers, including Hao Zhu as the Reviewing Editor and Reviewer #1, and the evaluation has been overseen by Sean Morrison as the Senior Editor.

The reviewers have discussed the reviews with one another and the Reviewing Editor has drafted this decision to help you prepare a revised submission.

Summary:

It is thought that NF-κB promotes inflammation and hepatocyte survival secondary to liver insults. That said, the effects of NF-κB on liver biology can depend on the cell type and the degree of NF-κB activation or inhibition, thus this is a complex but important area of study. It is somewhat surprising but clearly demonstrated in this manuscript that NIK, a kinase that is known to activate the non-canonical NF-κB pathway, is a suppressor of hepatocyte proliferation in both acute and chronic settings. The results presented in this manuscript are generally well performed and clearly presented, but additional experiments are needed to more rigorously support the conclusions of the paper. The genetic models support the notion that NIK-IKKα are negative regulators of hepatocyte proliferation in liver regeneration after partial hepatectomy and other injuries. However, some of the regeneration assays need to be more thoroughly performed and the epistatic relationships between NIK, IKKα, and JAK-Stat should be further elucidated. If these can be added to further support the conclusions of the paper, this story would add additional insight and nuance into traditional perspectives on the influence of NF-κB in liver disease.

Essential revisions:

1) Both reviewers 1 and 2 asked about the expression pattern of NIK in liver regeneration. Authors should provide a more detailed analysis of NIK before and after PHx (1 day, 2 day, 3 day, 5 day, 7 day) in wild-type mice. This would tell us whether or not expression of this gene is dynamically regulated during regeneration. It would also be nice if NIK expression before and after chronic injury in human livers could be more clearly shown or referenced.

2) The analysis of how loss of NIK affects liver regeneration should be more carefully performed. Authors should show a more detailed time course analysis on days 1, 2, 3, 5, and 7 to establish how NIK accelerates liver regeneration after PHx. Liver/body ratios should be reported for each time point. An analysis of Ki-67 at different time points would also provide a more complete picture of how and when NIK loss induces hepatocyte proliferation. In Figure 1 and in Figure 6, 7, The liver mass to body mass ratios should be reported for the pre-surgery mice, the resected liver pieces, and for the post-resection livers. This would represent the standard and more comprehensive way for how PHx liver regeneration phenotypes are generally characterized.

3) A more thorough protein expression analysis along the regeneration time course would help to illustrate how the major targets of NIK (JAK2, p-JAK2, STAT3, p-STAT3, IKKα) are affected in control and NIK KO mice. These expression studies should be performed in the same liver samples and multiple time points so as to help clarify the epistatic relationships in this signaling cascade.

4) The authors clearly show that IKKα deletion phenocopies NIK deletion, which suggests that IKKα is downstream of NIK. However in Figure 2, the authors claimed that phosphorylation of IKKα/β, IkBα, and p65 is not affected in NIK knockout mice compared to NIK floxed mice, therefore concluding that NF-κB pathway does not mediate the suppression of liver regeneration by hepatic NIK. Thus, two reviewers where not confident that the epistatic relationships between NIK, IKK, and Jak/Stat were conclusively established. Although liver-specific KO of NIK and IKKα show similar regeneration phenotypes, there is no data to directly assess the signaling cascade. This could be done by including NIK, p-IKKα and total-IKKα, together with p-STAT3/t-STAT3 and p-JAK/JAK in Western blot analyses in NIK KO, IKKα KO, and WT livers.

Along the same lines, the effect of NIK overexpression in IKKα-/- hepatocytes on JAK-STAT3 signaling should be examined to determine whether IKKα is a downstream effector of NIK. To limit the amount of work, this can be done in hepatocytes in vitro only and p-STAT3 can be used as a readout. If the time course signaling analysis within WT mice (comment 3) and these over-expression experiments clearly support the hypothesis that NIK is upstream of IKKα and Jak-stat, then the authors do not need to provide the signaling analysis in each of the different mutant mouse models.

5) The authors have published a paper in Endocrinology 2017 that states in the Abstract: "deletion of NIK in the liver, including both hepatocytes and immune cells, protected against HFD-induced liver steatosis and attenuated hepatic glucose production. Mechanistically, deletion of liver NIK suppressed liver inflammation and lipogenic programs, thus contributing to protection against liver steatosis." Can the authors discuss potential non-cell-autonomous effects that might accelerate hepatocyte regeneration? No experiments are necessary here, but a clarifying discussion about known cell-type dependent effects of NIK would be helpful to frame the findings in this paper in the context of past studies.

6) It was noted by reviewer 3 that the authors' claim that the described pathway regulates a quality control mechanism to block proliferation of damage hepatocytes is not supported by direct evidence. In fact, serum ALT, the only measurement that indirectly determines liver integrity, did not differ between control and liver NIK-/- mice. Please edit or clarify the manuscript to limit this particular statement.

[Editors' note: further revisions were requested prior to acceptance, as described below.]

Thank you for resubmitting your work entitled "Hepatic NF-κB-inducing Kinase (NIK) Suppresses Liver Regeneration in Mouse Acute and Chronic Liver Disease" for further consideration at *eLife*. Your revised article has been evaluated by Sean Morrison (Senior Editor), a Reviewing Editor, and two reviewers.

There have been some substantial improvements made during the revisions, especially in regards to the mechanistic relationships within the NIK IKK signalling axis. However, there are some significant concerns about the liver regeneration phenotypes that we would like you to clarify before we make a decision on this manuscript.

The liver regeneration data are heavily dependent on precise liver mass measurements in large numbers of mice at different time points. Liver mass recovery is ultimately the most important physiological readout (not necessarily proliferation) when assessing liver regeneration. With this in mind, the revision reveals significant uncertainty about the central phenotypic findings of the paper.

You present a method of calculation of "liver growth rates" during regeneration that is not accurate. You calculate liver growth rate as follows: "Estimation of total liver weight before PHx: resected liver weight ÷ 70%. Calculation of the remnant liver weight after PHx: total liver weight – resected liver weight. Liver weight gains: terminal liver weight – the remnant liver weight. Liver growth rates: liver weight gains normalized to the remnant liver weight after PHx."

The 70% number is theoretical, not measured, and each surgery can be different in terms of amount of liver resected and probably ranges from 50-75%. If in actuality less than 70% is resected, then the total liver mass will be substantially mis-calculated and under-estimated. This would be compounded when "liver growth rates" are calculated because the formulation stated above uses several non-measured numbers to calculate the remnant liver weight. This is likely why there is a significant increase in liver growth rates in Figure 1I, while there is no actual difference in liver/body weight ratio at 2 or 4 days as shown in Figure 1—figure supplement 1. The day 4 data does not contain enough mice and the p value does not reach significance. The negative data in this figure supplement is in opposition to Figure 1I, and suggests that there is not a significant effect on liver mass recovery after partial hepatectomy for NIK deleted mice.

The IKK data in Figure 4—figure supplement 1 is also confusing, with a significant differences in Ki67 but no strong changes in liver mass until day 7, when liver regeneration is mostly completed. The most rigorous data here are the liver/body weight ratios at days 2 and 4 for both NIK and IKK mice, and both of these sets of data are essentially negative.

Again in Figure 6E the same type of data discrepancy exists. In Figure 6E, there is a significant increase in liver growth rates, but then in the supplement there is no significant difference in liver/body weight ratios. All of these points raise the possibility that the phenotyping was not completely solid, and the revision experiments further bring those points to light. We would like to give you another opportunity to clarify the data and to resolve the questions raised above, if you are able to do so. An adequate number of mice to answer the questions would be important in this case. If the key data are indeed negative, or if the conclusions are not strongly supported by a rigorous way of assessing liver mass, then we would encourage you to resubmit this manuscript elsewhere.

---

## [Author Response]

Essential revisions:1) Both reviewers 1 and 2 asked about the expression pattern of NIK in liver regeneration. Authors should provide a more detailed analysis of NIK before and after PHx (1 day, 2 day, 3 day, 5 day, 7 day) in wild-type mice. This would tell us whether or not expression of this gene is dynamically regulated during regeneration. It would also be nice if NIK expression before and after chronic injury in human livers could be more clearly shown or referenced.

We conducted the requested experiments and added the new results in the revised Figure 2—figure supplement 1A. Indeed, we observed a transient increase in NIK expression, likely due to increased cytokine secretion and/or hepatocellular stress in response to PHx-induced liver injury. We tried to measure NIK protein levels using commercial anti-NIK antibodies from 3 sources (Santa Cruz sc-8417; Abcom ab191591, CST #4994), but failed to detect hepatic NIK protein. Notably, it is widely known that endogenous NIK levels in most animal tissues are below the detection thresholds of commercial antibodies.

We previously reported that liver NIK pathways are aberrantly activated in humans with chronic liver injury induced by alcoholic consumption or autoimmune attack (Hepatology, 60:2065-76, 2014). We cited this work in the revision.

2) The analysis of how loss of NIK affects liver regeneration should be more carefully performed. Authors should show a more detailed time course analysis on days 1, 2, 3, 5, and 7 to establish how NIK accelerates liver regeneration after PHx. Liver/body ratios should be reported for each time point. An analysis of Ki-67 at different time points would also provide a more complete picture of how and when NIK loss induces hepatocyte proliferation. In Figure 1 and in Figure 6, 7, The liver mass to body mass ratios should be reported for the pre-surgery mice, the resected liver pieces, and for the post-resection livers. This would represent the standard and more comprehensive way for how PHx liver regeneration phenotypes are generally characterized.

We performed the suggested experiments on NIK*^Δhep^* mice on day 2, 4, but not after day 4, post-PHx, because we were unable to obtain enough mice. We have encountered difficulty with NIK*^Δhep^* breeding colonies, and were unable to expand the colonies. Liver/body weight ratios were higher in NIK*^Δhep^* than in *NIK^f/f^*mice on day 4 post-PHx (the revised Figure 1—figure supplement 1).

As an alternative approach, we conducted the time course analyses of both liver/body weight ratios and Ki-67 cell number on IKKα*^Δhep^* mice from day 0 through day 7 following PHx. We found that liver/body ratios were significantly higher in IKKα*^Δhep^* than in *IKKα^f/f^*mice on day 7 post-PHx (the revised Figure 4—figure supplement 1B). We also assessed the time course of liver Ki67 cells (day 0-7) following PHx (Figure 4—figure supplement 1A).

We calculated liver/body ratios for Figures 1, 6 and 7 as suggested, and added resected liver/body ratios and post-resection liver/body ratios in the revised Figure 1—figure supplement 1 and Figure 6—figure supplement 1 (for Figures 6 and 7), respectively. We did not examine liver/body ratios in pre-surgery mice, because we do not have enough mice. Moreover, the results are unlikely to provide, in our view, significant new information, considering the results from the resected livers (resembling pre-surgical conditions).

3) A more thorough protein expression analysis along the regeneration time course would help to illustrate how the major targets of NIK (JAK2, p-JAK2, STAT3, p-STAT3, IKKα) are affected in control and NIK KO mice. These expression studies should be performed in the same liver samples and multiple time points so as to help clarify the epistatic relationships in this signaling cascade.

We carried out the requested experiments, and added new data in the revised Figure 2—figure supplement 1B-C (for IKKα) and the revised Figure 5—figure supplement 1A-B (for JAK2 and STAT3). We discussed these experiments in the following response to Question 4.

4) The authors clearly show that IKKα deletion phenocopies NIK deletion, which suggests that IKKα is downstream of NIK. However in Figure 2, the authors claimed that phosphorylation of IKKα/β, IkBα, and p65 is not affected in NIK knockout mice compared to NIK floxed mice, therefore concluding that NF-κB pathway does not mediate the suppression of liver regeneration by hepatic NIK. Thus, two reviewers where not confident that the epistatic relationships between NIK, IKK, and Jak/Stat were conclusively established. Although liver-specific KO of NIK and IKKα show similar regeneration phenotypes, there is no data to directly assess the signaling cascade. This could be done by including NIK, p-IKKα and total-IKKα, together with p-STAT3/t-STAT3 and p-JAK/JAK in Western blot analyses in NIK KO, IKKα KO, and WT livers.

We carried out the requested experiments, and added new data in the revised Figure 2—figure supplement 1B-C (IKKα) and Figure 5—figure supplement 1A-B (JAK2 and STAT3). Phosphorylation of IKKα appeared to precede a decrease in phosphorylation of JAK2 in wild type mice following PHx; deletion of hepatic *IKKα* largely blocked reduction in JAK2 phosphorylation (Figure 5—figure supplement 1B). These data further support the notion that hepatic IKKα suppresses JAK2 following PHx.

Along the same lines, the effect of NIK overexpression in IKKα^-/-^ hepatocytes on JAK-STAT3 signaling should be examined to determine whether IKKα is a downstream effector of NIK. To limit the amount of work, this can be done in hepatocytes in vitro only and p-STAT3 can be used as a readout. If the time course signaling analysis within WT mice (comment 3) and these over-expression experiments clearly support the hypothesis that NIK is upstream of IKKα and Jak-stat, then the authors do not need to provide the signaling analysis in each of the different mutant mouse models.

We carried out the suggested hepatocyte culture experiments and added new data in the revised Figure 5—figure supplement 1C-D. IKKα deficiency decreased, but not completely blocked, NIK-induced suppression of phosphorylation of STAT3 in response to IL6. These new data further support the original hypothesis that hepatic NIK is able to suppress liver regeneration by both IKKα-dependent and –independent mechanisms.

5) The authors have published a paper in Endocrinology 2017 that states in the Abstract: "deletion of NIK in the liver, including both hepatocytes and immune cells, protected against HFD-induced liver steatosis and attenuated hepatic glucose production. Mechanistically, deletion of liver NIK suppressed liver inflammation and lipogenic programs, thus contributing to protection against liver steatosis." Can the authors discuss potential non-cell-autonomous effects that might accelerate hepatocyte regeneration? No experiments are necessary here, but a clarifying discussion about known cell-type dependent effects of NIK would be helpful to frame the findings in this paper in the context of past studies.

We expanded Discussion as requested, adding “Notably, deletion of hepatocyte *NIK* increased phosphorylation of hepatic JAK2 and STAT3 in NIK*^Δhep^* mice either treated with AAF or fed a HFD, suggesting that hepatic NIK suppresses hepatocyte replication in part by repressing the JAK2/STAT3 pathway. Notably, NIK activity in nonparenchymal cells, including immune cells, was recently reported to contribute to obesity-associated liver steatosis (27). We propose that in chronic liver disease, NIK pathways in nonparenchymal cells, particularly in Kupffer cells/macrophages, may also inhibit reparative hepatocyte replication indirectly by a paracrine mechanism. Relative contributions of the hepatocyte-autonomous and non-autonomous actions of NIK to liver regeneration should be further studied in the future”.

6) It was noted by reviewer 3 that the authors' claim that the described pathway regulates a quality control mechanism to block proliferation of damage hepatocytes is not supported by direct evidence. In fact, serum ALT, the only measurement that indirectly determines liver integrity, did not differ between control and liver NIK-/- mice. Please edit or clarify the manuscript to limit this particular statement.

We deleted the quality control-related discussion in revision as suggested.

[Editors' note: further revisions were requested prior to acceptance, as described below.]

The liver regeneration data are heavily dependent on precise liver mass measurements in large numbers of mice at different time points. Liver mass recovery is ultimately the most important physiological readout (not necessarily proliferation) when assessing liver regeneration. With this in mind, the revision reveals significant uncertainty about the central phenotypic findings of the paper.You present a method of calculation of "liver growth rates" during regeneration that is not accurate. You calculate liver growth rate as follows: "Estimation of total liver weight before PHx: resected liver weight ÷ 70%. Calculation of the remnant liver weight after PHx: total liver weight – resected liver weight. Liver weight gains: terminal liver weight – the remnant liver weight. Liver growth rates: liver weight gains normalized to the remnant liver weight after PHx."The 70% number is theoretical, not measured, and each surgery can be different in terms of amount of liver resected and probably ranges from 50-75%. If in actuality less than 70% is resected, then the total liver mass will be substantially mis-calculated and under-estimated. This would be compounded when "liver growth rates" are calculated because the formulation stated above uses several non-measured numbers to calculate the remnant liver weight. This is likely why there is a significant increase in liver growth rates in Figure 1I, while there is no actual difference in liver/body weight ratio at 2 or 4 days as shown in Figure 1—figure supplement 1. The day 4 data does not contain enough mice and the p value does not reach significance. The negative data in this figure supplement is in opposition to Figure 1I, and suggests that there is not a significant effect on liver mass recovery after partial hepatectomy for NIK deleted mice.

We thank the reviewers and editors for these constructive comments. We deleted the “liver growth rates” data following these comments. We performed additional PHx experiments, increasing animal number at 4 days from 4 to 8 mice per group. Excitingly, we observed that liver/body weight ratios are significantly higher in *NIK^Δhep^*than in *NIK^f/f^*mice. We replaced the original Figure 1I with these new results in the revised Figure 1I.

Of note, there is no significant difference in liver/body weight ratios at 2 days post-PHx. We reason that a 2-day period is too short for hepatocytes to grow in size to significantly increase liver mass and liver/body weight ratios. In this short initial phase, hepatocytes rapid replicate, but newly-generated hepatocytes are not yet grow in size to increase total liver mass. Hepatocyte hypertrophy likely occurs later, thereby increasing liver weight and liver/body weight ratios at relatively late stages (i.e. >3 days). We provided this explanation in the revised manuscript.

The IKK data in Figure 4—figure supplement 1 is also confusing, with a significant differences in Ki67 but no strong changes in liver mass until day 7, when liver regeneration is mostly completed. The most rigorous data here are the liver/body weight ratios at days 2 and 4 for both NIK and IKK mice, and both of these sets of data are essentially negative.

We conducted additional PHx experiments, increasing mouse number: day 0 and 1: from 3 to 4 per group; day 3: *IKKα^f/f^*: 5 to 6, *IKKα^Δhep^*: 5 to 8; day 5: *IKKα^f/f^*: 5 to 9, *IKKα^Δhep^*: 5 to 8; day 7: *IKKα^f/f^*: 5 to 6. We found that liver/body weight ratios are significantly higher in *IKKα^Δhep^*than in *IKKα^f/f^*mice at both 5 and 7 days post-PHx. They become higher at 3 days, but not statistically different. We replaced the original Figure 4F (liver growth rates) with these new results in the revised Figure 4F. We also provided an explanation that a short duration of the initial phase (<3 days post-PHx) likely limits the growth of newly-generated hepatocytes, contributing to modest liver/body weight ratio difference at this stage.

Again in Figure 6E the same type of data discrepancy exists. In Figure 6E, there is a significant increase in liver growth rates, but then in the supplement there is no significant difference in liver/body weight ratios.

We deleted Figure 6E (liver growth rates) following the comments discussed in the Response 1. The liver/body weight ratio experiments were conducted at 2 days post-PHx in both Figures 6 and 7. Based on the above discussions about the revised Figures 1I and 4F, liver/body weight ratios are expected to be comparable between *NIK^Δhep^*than in *NIK^f/f^*mice at 2 days, because this short duration limits growth of newly-generated hepatocytes and liver mass. We provide this explanation in the revise manuscript.